# Structural FEA-Based Design and Functionality Verification Methodology of Energy-Storing-and-Releasing Prosthetic Feet

**Johnnidel Tabucol** [1,2,3,*] **, Tommaso Maria Brugo** [1,2] **, Marco Povolo** [1] **, Marco Leopaldi** [2] **, Magnus Oddsson** [4] **, Raffaella Carloni** [3] **and Andrea Zucchelli** [1,2,*]

1 Department of Industrial Engineering, University of Bologna, 40131 Bologna, Italy; tommasomaria.brugo@unibo.it (T.M.B.); marco.povolo2@unibo.it (M.P.)
2 Interdepartmental Centre for Industrial Research in Advanced Mechanical Engineering Applications and Materials Technology, University of Bologna, 40131 Bologna, Italy; marco.leopaldi@unibo.it
3 Bernoulli Institute for Mathematics, Computer Science and Artificial Intelligence, Faculty of Science and Engineering, University of Groningen, 9747 AG Groningen, The Netherlands; r.carloni@rug.nl
4 Research and Development, Össur, 110 Reykjavík, Iceland; magnuso@ossur.com
* Correspondence: johnnidel.tabucol2@unibo.it (J.T.); a.zucchelli@unibo.it (A.Z.); Tel.: +39-320-8177937 (J.T.); +39-348-4559575 (A.Z.)

**Abstract:** The prosthetic feet that are most often prescribed to individuals with K3/K4 levels of ambulation are the ESR feet. ESR stands for energy-storing and -releasing. The elastic energy is stored by the elastic elements in composite materials (carbon fiber or glass fiber). ESR feet must be developed and optimized in terms of stiffness, taking into account the loads that a healthy human foot undergoes and its kinematics while walking. So far, state-of-the-art analyses show that the literature approaches for prosthetic foot design are not based on a systematic methodology. With the aim of optimizing the stiffness of ESR feet following a methodological procedure, a methodology based on finite element structural analysis, standard static testing (ISO 10328) and functional verification was optimized and it is presented in this paper. During the path of optimization of the foot prototypes, this methodology was validated experimentally. It includes the following: (i) geometry optimization through two-dimensional finite element analysis; (ii) material properties optimization through three-dimensional finite element analysis; (iii) validation test on physical prototypes; (iv) functionality verification through dynamic finite element analysis. The design and functional verification of MyFlex-$\gamma$, a three-blade ESR foot prosthesis, is presented to describe the methodology and demonstrate its usability.

**Keywords:** finite element analysis (FEA); prosthetic feet; stiffness optimization; biomechanics

## 1. Introduction

Current commercially available prostheses are mostly energy-storing -and -releasing (ESR) feet and they are the most prescribed prosthetic feet for individuals with K3 and K4 levels of ambulation. ESR feet are passive prosthetic devices made of elastic elements, which ensure the ESR feet work as springs that store energy during the mid-stance of the gait cycle and release it for the propulsion during late stance [1,2]. The elastic elements are generally leaf springs, also called blades, of composite materials (carbon fiber-reinforced plastic—CFRP; or glass fiber-reinforced plastic—GFRP). The stiffness of the elastic elements is a crucial characteristic of foot prostheses. Stiffness depends on the geometries and the material properties, especially for ESR feet [3,4], and the choice of the global stiffness category of foot prostheses depends on the weight and the users' activity levels [5]. Moreover, the few active ankle prosthetic devices on the market are joined to feet by a system of composite leaf springs (see Table 1 for remarkable examples). The contribution of the elastic elements is always significant.

The present work aims to propose and apply a systematic methodology that includes elastic elements in the design process of prosthetic feet. The methodology combines

numerical and experimental techniques for the design and the functional verification of foot prostheses. The proposed methodology consists of three main phases: (i) the *design phase*, which includes the stiffness and safety optimization; (ii) the *mechanical test phase* on a physical prototype; and (iii) the *functionality verification phase*. The stiffness optimization, carried out during the *design phase*, is performed using static structural finite element (FE) analyses (FEA). Optimizing a passive foot prosthesis means optimizing its stiffness in such a way that it gives rotations in both dorsiflexion and plantarflexion similar to the rotations of a healthy foot. In the ankle–foot system of a human who has not suffered any amputation to the lower limbs, there are muscles that control every movement giving the desired or necessary rotation. With transtibial amputation, most of the muscles are eliminated or reduced and a passive device such as the ESR foot cannot guarantee active control as muscles do. However, by studying the literature, it is possible to determine the loads to which a healthy foot is subjected while walking and its corresponding rotations. Therefore, an ESR foot can be optimized in terms of stiffness considering the loads it is subjected to and the rotations it must make while walking.

**Table 1.** Commercial active and semi-active foot prostheses with composite elastic foot.

| Manufacturer | Model | Website (Access Date) | Country |
|---|---|---|---|
| Blatchford | Elan | www.blatchford.co.uk (1 November 2021) | UK |
| Blatchford | Elan$^{IC}$ | www.blatchford.co.uk (1 November 2021) | UK |
| Fillauer | Raize | www.fillauer.com (1 November 2021) | USA |
| Freedom-Innovations | Kinnex 2.0 | www.freedom-innovations.com (1 November 2021) | USA |
| Össur | Proprio Foot | www.ossur.com (1 November 2021) | Iceland |
| Ottobock | Empower | www.ottobock.com (1 November 2021) | Germany |

In previous publications, FEA was used with different aims in the design/study of prosthetic feet. Omasta et al. and Bonnet et al. used FEA to analyze the stress-strain behavior of load-bearing components [6,7]. Bonnet et al. also used FEA to analyze the energy stored by an ESR foot [7]. FEAs were also applied to the design of foot prostheses that comply with standards such as ISO 22675 (www.iso.org, accessed on 1 January 2021) [8] or *American Orthotic and Prosthetic Guidelines* (AOPA, www.aopanet.org, accessed on 1 January 2021) [9]. FEA was used by Prost et al. to carry out the stiffness optimization of the elastic elements of passive [10] prosthetic feet, while Sheperd et al. used it for the same aim but to optimize a quasi-passive [11] foot prosthesis. The profile shapes of prosthetic feet were to obtain an optimal roll-over by carrying structural FEAs [12,13]. Dao et al. designed an ESR foot with elastic elements in the composite materials (glass fiber) [14]. Rigney et al. included FEA in a methodology that concerned ESR prosthetic foot characterization; however, they did not propose it as a tool to design a new energy-storing and -releasing foot [15]. Tryggvason et al. presented a work where the main aim was to create a model which was meant to serve as a platform for iterative modifications of a foot prosthesis design by simulating a standardized dynamic mechanical test (ISO/TS 16955), wherein the foot performed a complete roll-over [16]. Table 2 summarizes the literature on the use of FEA for the design and analysis of prosthetic feet characterized by the elastic elements in composite materials. So far, state-of-the-art analyses show that the literature approaches relatively to prosthetic foot design are not based on a systematic methodology.

This work aims to introduce a methodology that helps the designer develop their initial idea of a foot prosthesis by combining ergonomic and functional requirements with stiffness and strength requirements and choosing the most suitable materials. In the methodology, two different FE models of the foot prosthesis were built during the first phase (*design phase*, Section 3.1). The first FE model was two-dimensional (2D FE model) and considered the foot prosthesis only in the sagittal plane (Section 3.1.1). The second FE model was three-dimensional (3D FE model); the profile shapes of the elastic elements in the sagittal plane were the ones optimized in the 2D FE analysis (Section 3.1.2). In previously published studies, 3D FEAs have been carried out to perform stiffness optimization of the energy-storing parts [12,13,16]. However, when the analysis aims to study the effect of

varying many geometric parameters on the behavior of the prosthesis, 3D FEA is time-consuming. Therefore, the 2D FE model built in the present work is aimed to be used as a tool to determine the effect of the geometry variation on the foot prosthesis behavior in the sagittal plane, reducing the computation time. The 3D FE model was built to perform a more detailed structural analysis, aiming to optimize the stiffness and the strength of the prosthetic device. The optimization was carried out by determining the final material properties to use; therefore, the lamination sequence of the layers of CFRP was used to build the elastic elements. In previous studies, the composite elastic elements have been simplified by simulating them with isotropic properties in 3D FEAs [12–14]. Simplifying the laminate composites as isotropic materials could lead to high approximated results, both in terms of stiffness and strength points of view. In Section 3.1.2, a guideline to build the 3D FE model is given, also specifying the constraints and the load conditions of the equivalent of ISO 10328 static tests. The second phase of the present methodology is the FE model validations. The 2D and 3D FE models and analyses were subsequently subjected to validation through an ISO 10328-inspired static test (*mechanical test phase*, see Section 3.2). The *design phase* was carried out by optimizing the stiffness of the ESR foot with static loads; thus, the *validation phase* was carried out with static tests. Nevertheless, the foot prostheses were loaded dynamically during use. Therefore, it is crucial to understand the behavior of the foot prostheses when they are under dynamic conditions. Two approaches to verify the functionality and study prosthetic feet's behavior when subjected to dynamic loads are proposed. They were both carried out through two-dimensional FE analysis, wherein only the motions in the sagittal plane were considered. The load conditions of the first approach consisted of a simplified version of the dynamic test proposed in the ISO 10328 standard, whereby the foot was loaded at the heel and toe with two platforms, simulating the ground reaction forces (Section 3.3.2). The second approach is based on the ISO 22675 and ISO/TS 16955 standards load conditions, whereby a tilting table simulated the relative rotation between the thigh and the ground and an actuator simulated the ground reaction forces by pushing the thigh–shank–foot system (Section 3.3.3) downwards. As a case study to describe the methodology and demonstrate its usability, the stiffness optimization of an ESR foot is presented: the MyFlex-$\gamma$ foot, size 25, for 60 kg users and K3/K4 as a level of ambulation. The configuration of the composite blades of MyFlex-$\gamma$ is similar to the ESR foot Pro-Flex Pivot by Össur.

**Table 2.** Aim and type of simulations and material properties used in previous works where finite element analysis is applied to study foot prostheses.

| Ref. | Aim | Type | Mat. Prop. |
| --- | --- | --- | --- |
| Omasta et al. [6] | analysis | 3D Static | linear, isotropic |
| Bonnet et al. [7] | analysis | 3D static | linear, isotropic |
| Naveed et al. [8] | design | 3D dynamic | linear, isotropic |
| Santana et al. [9] | design | 3D static | non linear, orthotropic |
| Prost et al. [10] | design | 3D static | linear, isotropic |
| Shepherd et al. [11] | design | 3D static | linear, isotropic |
| Mahmoodi et al. [12] | design | 3D dynamic | linear, isotropic |
| Ke et al. [13] | design | 3D static | linear, isotropic |
| Dao et al. [14] | design | 3D static/dynamic | linear, isotropic |
| Rigney et al. [15] | analysis | 3D static/dynamic | linear, isotropic |
| Tryggvason et al. [16] | design | 3D dynamic | non linear, orthotropic |

## 2. Requirements

Designing ESR feet means optimizing the stiffness of their elastic elements. The basic definition of stiffness of a loaded structure is *stiffness* equal to *load* divided by the *deformation*. In the present case, the deformation corresponds to the rotation of the foot in the sagittal plane (dorsiflexion and plantarflexion). On the other hand, the load corresponds to the ground reaction forces. In this section, the ankle–foot range of rotation on the sagittal plane (Section 2.1) and the *ground reaction forces* (Section 2.2) during *normal ground walking* are

given. *Normal ground walking* is a cyclic movement. The cyclic movement is properly called the gait cycle or stride. It consists of two main phases, the *stance phase* (when the foot is touching the ground) and the *swing phase*. For the aim of the design methodology, the work is focused on the *stance phase* only. The *stance phase* consists of three sub-phases: the *early stance*, the *mid stance* and the *late stance*. The heel portion of the ESR foot absorbs the impact of the foot prosthesis against the ground in the *early stance*; the elastic elements store elastic energy during *mid stance*, then release it during the *late stance* for the *push off*.

## 2.1. The Ankle Rotation on the Sagittal Plane

The ankle range of motion while walking on normal ground is significantly influenced by several factors, such as walking velocity, age and gender [17–24]. However, the curves that describe the behavior of the ankle rotations for different conditions follow the same path. Thus, the healthy human ankle rotation has an initial plantarflexion during *early stance* (from 8% to 12% of the gait cycle) that goes from $-1°$ to $-8°$. The maximum dorsiflexion before *heel-off* occurs from 60% to 70% of the gait cycle and goes from 6° to 16° of ankle rotation.

## 2.2. The Vertical Ground Reaction Forces during Normal Ground Walking

The gait cycle is influenced by different factors, not only kinematically but also in terms of how the foot impacts the ground at *heel strike* and *toe strike* and how it loads the ground during the entire *stance phase*. The forces exchanged by the foot and the ground are called *ground reaction forces*. The value of *ground reaction forces* during normal walking vary under different conditions, for instance, depending on different walking velocities [21]. The vertical *ground reaction forces* during normal ground walking have two peaks (*M-shaped ground reaction force*); the first peak is at the *early stance* and it is between 95% to 130% of the body weight; the second peak is at the end of the *mid-stance* and it is from 95% to 105% of the body weight.

## 2.3. The ISO 10328 Standard Static Test

The stiffness characterization and safety verification of ESR feet and foot prostheses, in general, are carried out with static tests regulated by standards, such as the ISO 10328, or the *AOPA Guidelines*. According to both the standard and the guidelines, the forefoot and the heel are loaded in two separate tests. The force is imposed on the forefoot and the heel with a platform. For the heel test/plantarflexion test, the foot is relatively inclined backwards by 15° to the platform to simulate the angle between the foot and the ground at heel strike. When loaded, the foot prosthesis is subjected to a plantarflexion. For the toe test/dorsiflexion test, the foot is relatively inclined forward by 20° to the platform to simulate the angle between the foot and the ground at the end of the stance phase, at the toe-off. When loaded, the prosthetic device is subjected to dorsiflexion. For the stiffness determination of the foot prosthesis, the *settling force* (ISO 10328) is used and its maximum value depends on the weight category of the intended users. For the plantarflexion test, the settling force at the heel is from 105% (100 kg weight category) to 125% (60 kg weight category) of the body weight of the user. For the dorsiflexion test, the settling force at the forefoot is from 94% (100 kg user) to 108% (60 kg user) of the body weight of the user. Both the forces are imposed with a rate between 100 N/s and 250 N/s.

## 2.4. Biomechanical Requirements

Based on the kinematics (Section 2.1), the loads (Section 2.2) and the standard test (Section 2.3), the target is to optimize the elastic elements of the foot prosthesis in order to provide the aimed rotations both in plantarflexion and dorsiflexion during normal ground walking; as objective chosen for the present work, the foot must have a plantarflexion at *early stance* comprised in the range between $-5°$ and $-8°$ when the heel is loaded with between 95% and 130% of the body weight of the intended users and dorsiflexion in the range between 14° and 18° when the forefoot is loaded between around 95% and 108%

of the weight category. The values of dorsiflexion and plantarflexion and of the loads corresponding to them vary according to the end user; therefore, even if using this same methodology, the biomechanical objectives can vary.

### 2.5. Foot Prosthesis Configuration

The ESR foot considered to demonstrate the usability of the design methodology is MyFlex-γ, whose design is inspired by Pro-Flex Pivot by Össur. MyFlex-γ can be subdivided into three functional groups (see Figure 1), i.e., the ankle group (ankle frame and tube connector), the tendon group (link and spring holder) and the foot group (lower blade, middle blade and upper blade).

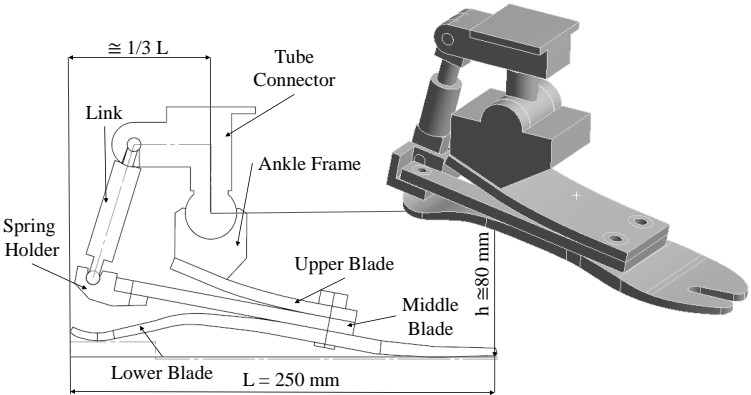

**Figure 1.** Dimensions and simplified 3D CAD of MyFlex-γ.

When the foot is loaded at the heel (*early stance*, before *toe strike*), the heel portion of the lower blade, as well as the middle blade, is subjected to a deflection. On the other hand, when the foot is loaded at the forefoot (*late stance*, after *heel-off*), the upper blade and the middle blade are deflected and deformed in respect to each other—see Figure 2. The deflection of each of the above-mentioned elastic parts depends on their elasticity, intended as material properties, and their geometries, which define the contact—thus, how they deflect each other. Given this, the geometry and material optimizations are fundamental to reaching the prosthetic foot's targeted stiffness.

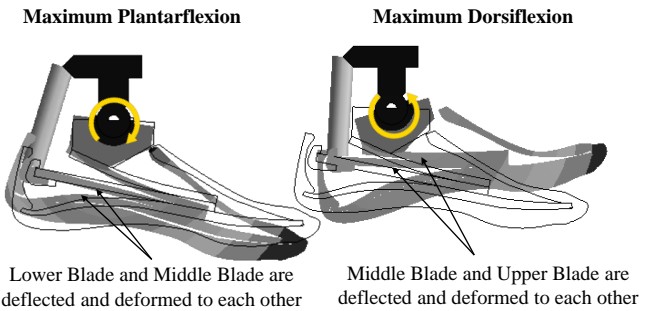

**Figure 2.** Deflection of the blades in respect to each other when the foot prosthesis is loaded at the heel and at the forefoot.

## 3. Materials and Method

The design and functionality verification methodology proposed consists of a procedure made of three main phases: (i) the *design phase*, (ii) the *mechanical test phase* to validate the *design* and (iii) the *functional verification phases*. In Figure 3, the flowchart of the design and functionality verification is depicted.

Once the requirements were defined and once the foot prosthesis configuration was chosen, an initial geometry of the foot prosthesis (two-dimensional CAD model) was drawn

and used in the *design phase*. The FE models built and used to optimize the stiffness of the foot prosthesis were then validated through static tests, loading a physical prototype of the prosthesis. Functional verification was subsequently made through two 2D dynamic FEAs built upon the 2D FE model of the *design phase*.

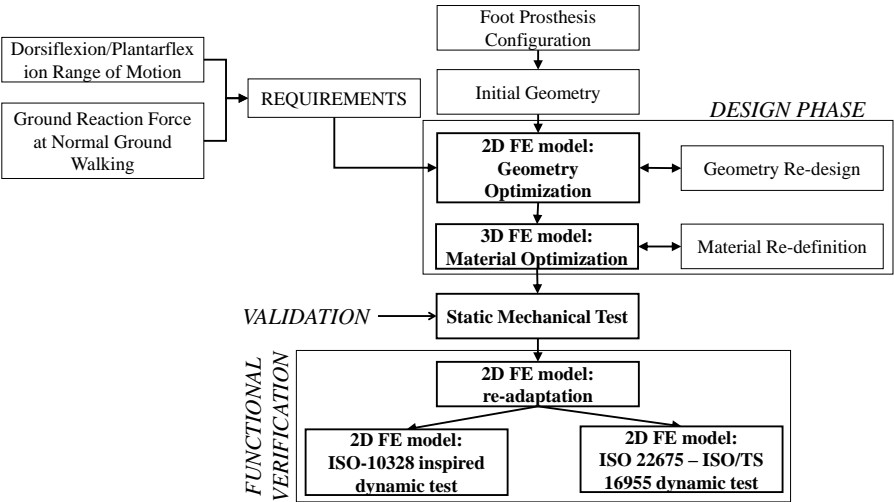

**Figure 3.** Flowchart of the design methodology: *design phase*, *validation phase* and *functional verification phase*.

### 3.1. Design Phase

The most important locomotion's degree of freedom of the foot is the rotation in the sagittal plane. Therefore, to simplify the preliminary stiffness optimization of the prosthesis, based mainly on the modification of the geometry of the prosthetic device, a 2D FE model was built to simulate the motion of the foot in the sagittal plane (Section 3.1.1). Once the geometry in the sagittal plane was defined, a 3D FE model was built to carry out a more detailed simulation, wherein the composite elastic elements were optimized by modifying the laminate material properties, working mainly on the type, orientation and numbers of layers of CFRP prepregs (Section 3.1.2). Before explaining in detail the various stages of the proposed methodology, it is reminded that optimizing a passive foot prosthesis means optimizing its stiffness in such a way that it gives rotations in both dorsiflexion and plantarflexion similar to the rotations of a healthy foot, when such prosthesis is subjected to loads (ground reaction forces) similar to those to which a healthy foot is subjected. For the present application, as already mentioned in Section 2.4, the aim is to optimize the prosthesis' stiffness in such a way that, when it is loaded at the heel with a load between 95% and 130% of the weight of the user, it gives a plantarflexion between −5° and −8°; when it is loaded at the toe with a load between 95% and 108% of the body weight, it must give a dorsiflexion between 14° and 18°.

#### 3.1.1. Geometry Optimization: 2D FE Model

In this first step of the *design phase*, geometry optimization was carried out through a 2D FEA. For example, this step can be used to optimize the profile shapes of an already defined configuration of elastic elements, to configure the energy-storing parts of a new ESR foot, or to investigate the behavior of an existing prosthetic device after modifications or additions of other functional components, such as dampers and actuators.

Optimizing the geometry of a foot prosthesis is very important, especially with regard to ESR feet and their elastic components. Although the result of the 2D FE model in terms of stiffness is not definitive, this model is essential to optimize the shapes of the various elastic components to ensure the full range of motion needed to walk. A non-optimization of the shapes of the elastic components can bring a range of motion too reduced or too wide. Regardless of the stiffness of the elastic components, if the range of motion is too

small, the foot rotates around the ankle joint up to a certain angle and then, a sudden sensation of increase in the equivalent rotational stiffness around the ankle joint follows. Due to this sudden increase in stiffness of the ankle joint, the foot no longer rotates around the ankle and the heel comes off the ground before the desired moment. This situation can be perceived by the user as a sudden disappearance of the support, which can cause discomfort or even a fall of the user. If the range of motion is too wide, the foot may continue to rotate in dorsiflexion beyond the desired limit by delaying the hell-off. Even this situation can generate discomfort or even a fall of the user.

For MyFlex-$\gamma$, geometry optimization was focused on the profile shape of the middle blade and the upper blade, aiming to meet the biomechanical requirements previously defined (Section 2.4). The lower blade used for the present application was provided by Össur and it was already optimized for the chosen weight category. Then, an initial 2D geometry of the foot was drawn bearing in mind the dimensional parameters defined by the human anatomy such as length $L$ of the foot, the height $h$ of the ankle joint from the ground and the distance $d$ of the ankle joint from the heel, which is around 1/3 of the foot length. For the present application, the length of the foot was 250 mm and the height of the ankle joint was between 80 mm and 100 mm, as shown in Figure 1.

*Load conditions*. The CAD model was drawn in the $x$–$y$ plane and imported into ANSYS Workbench. The FE model was simulated in a 2D static structural analysis. Following the ISO 10328 standard, the dorsiflexion test was simulated by loading the FE model with a platform at the forefoot. For the plantarflexion test, the FE model was loaded at the heel. The platforms were moved by imposing 10 mm of displacement for the plantarflexion test and 50 mm for the dorsiflexion test. The shank was constrained with a fixed support at the top of the tube connector. For the present application, the loads and the constraints are depicted in Figure 4.

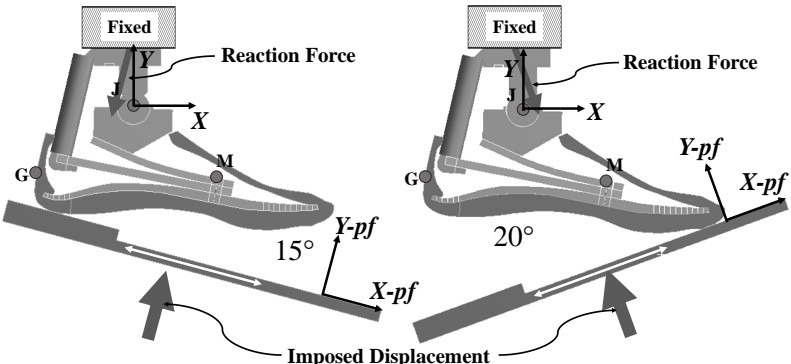

**Figure 4.** Schematic image of the boundary conditions used for the 2D FE model of the *Design* step, which was used to simulate the static tests according to ISO 10328. The heel was loaded with an inclined platform ($-15°$) to which a displacement was imposed (10 mm); the forefoot was compressed with a 20° inclined platform which was moved until 50 mm. In both cases, the platform was free to move along its longitudinal direction.

The geometry optimization was carried out by varying the geometric parameters. These parameters depend on the foot prosthesis architecture and the designers can choose them. For MyFlex-$\gamma$, the parameters are depicted in Figure 5 and listed in Table 3. The upper blade was defined in the sagittal plane by 6 parameters. The parameter $UB_t$ is the thickness, while the parameters $c_1$, $c_2$, $c_3$, $c_4$ and $c_5$ define the curvatures of the curved profile of the upper blade, which was composed by 6 straight sections—starting from the metatarsus (front), $c_1$ is the relative inclination between the first and the second section, $c_2$ is the relative inclination between the second and third section, etc. The middle blade, which had a straight profile in the sagittal plane, was defined only by its length ($MB_L$) and thickness ($MB_t$). The lower blade, provided by Össur, had been already optimized, in terms of shape and material properties, for specific users' weight categories.

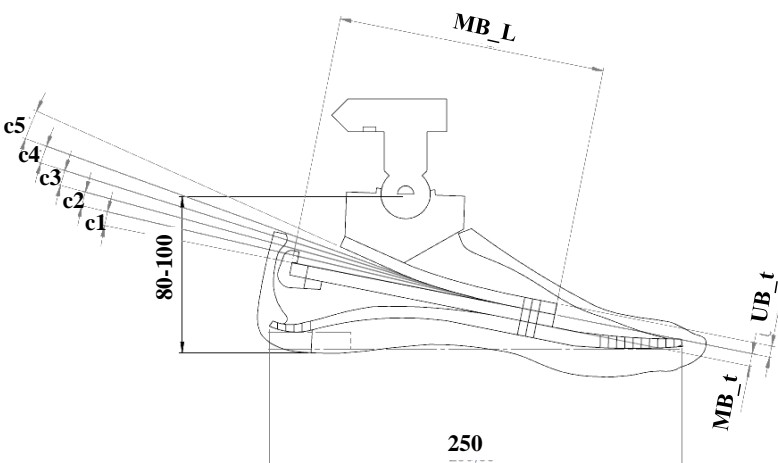

**Figure 5.** Geometric parameters of MyFlex-γ varied in the 2D FE model of the *design phase*—see Table 3.

**Table 3.** Geometric parameters of MyFlex-γ varied in the 2D FE model of the *design phase*—see Figure 5.

| Parameter | Min Value | Max Value | |
|---|---|---|---|
| upper blade thickness $UB_t$ (mm) | 6.00 | 8.50 | defined in step 2 |
| upper blade curvature 1 UB $c_1$ (deg) | 1.00 | 3.00 | |
| upper blade curvature 2 UB $c_2$ (deg) | 1.00 | 3.00 | |
| upper blade curvature 3 UB $c_3$ (deg) | 1.00 | 3.00 | |
| upper blade curvature 4 UB $c_4$ (deg) | 1.00 | 3.00 | |
| upper blade curvature 5 UB $c_5$ (deg) | 3.00 | 5.00 | |
| middle blade length $MB_L$ (mm) | 150 | 175 | |
| middle blade thickness $MB_t$ (mm) | 7.00 | 10.00 | defined in step 2 |

*Mesh modeling*. The FE model was built in ANSYS Workbench. PLANE183 (ANSYS) elements were used for mesh building; these were two-dimensional 8- and 6-node elements with quadratic displacement behavior and two translations at each node as degrees of freedom. The final 2D FE model of MyFlex-γ was meshed (Figure 6), presenting a total of 10,000 nodes corresponding to 20,000 degrees of freedom.

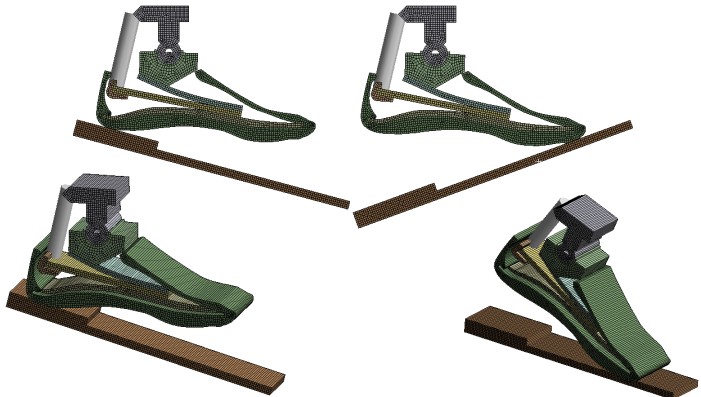

**Figure 6.** Mesh modeling and width assignment in the transversal direction. The 2D FE model was meshed with PLANE183 elements, for a total of 10,000 nodes without the platform and 20,000 with the platform.

*Contacts modeling*. The contacts between parts had to be modeled differently according to the real conditions. For parts that worked bending toward each other with slight sliding, the contacts were modeled as *frictional* using *pure penalty formulation* and 0.1 was set as

the *normal stiffness factor*. For contacts that could be modeled as glued/bonded, the *bonded* contact type was used with *augmented Lagrange formulation* and 1 was set as the *normal stiffness factor*.

In the present work, the upper blade, middle blade and lower blade bent toward each other with slight sliding when the foot was loaded. Therefore, the contacts between the upper blade and middle blade and between the middle blade and the lower blade were modeled as *frictional contact* using *pure penalty formulation*. The ankle frame and the upper blade were joined together with two M8 bolts along the longitudinal direction; for this condition, the contact between the ankle frame bottom surface and the upper blade top surface could be modeled as *bonded* to simplify the simulation, using *augmented Lagrange formulation*. A summary of the contacts is presented in Table 4.

**Table 4.** Contacts' properties. See also Figure 7. AF = ankle frame; UB = upper blade; MB = middle blade; LB = lower blade; SH = spring holder; TC = tube connector.

| Surface 1 | Surface 2 | Type | Formulation | Frict. Coeff. | Norm. Stiff. Fact. |
|-----------|-----------|------|-------------|---------------|--------------------|
| AF top | UB bottom | bonded | augm.Lagrange | - | 1.00 |
| UB bottom | MB top | frictional | pure penalty | 0.20 | 0.01 |
| MB bottom | LB top | frictional | pure penalty | 0.20 | 0.01 |
| MB bottom | SH top | frictional | pure penalty | 0.20 | 0.01 |
| AF ankle | TC ankle | no separation | augm.Lagrange | - | 1.00 |

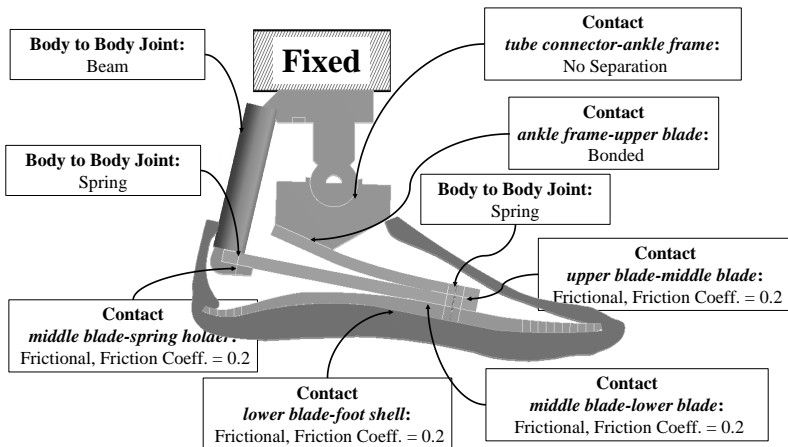

**Figure 7.** Joint and contact modeling in the 2D FE model. See also Table 4 for detailed information.

For the 2D FE model of MyFlex-$\gamma$, a *no-separation* contact type between the tube connector and the ankle frame (see Figure 7) was used to model the ankle joint; a *no-separation* contact type allows frictionless motions to be performed without separation of the parts.

*Joints*: Pretensioned bolts were modeled as preloaded springs, where the preload given as force was equal to the bolt pretension of the corresponding bolt. More specifically, the bolts were modeled as *longitudinal springs body-to-body joints* (longitudinal COMBIN14, ANSYS). Both the nodes of the COMBIN14 element were applied as *direct attachment* to the nodes of the connected parts. The elastic elements were joined together with two pretensioned M8 bolts in the physical MyFlex-$\gamma$ that could be seen as a single bolt in the sagittal plane. In addition, the middle blade and the spring holder were joined with preloaded M6 screws. Bolt pretension was simulated as a normal force given to the *longitudinal springs body-to-body joints*, which corresponded to the standard bolt pretensions for M6 (7.4 kN, 8.8 class, frictional coefficient = 0.20) and M8 (13.7 kN, 8.8 class, frictional coefficient = 0.20). Connection links with hinge joints at both extremities were modeled with *body-to-body beam joints* (BEAM3, ANSYS). BEAM3 is a 2D uniaxial element with tension, compression and bending capabilities and it has the translations in both directions

of the $x$–$y$ plane and the rotation around the z-direction at each node. In MyFlex-$\gamma$, the link part, characterized by two hinge joints that connected the tube connector to the spring holder, was modeled by BEAM3.

*Simulation conditions*: ESR feet elastic elements are subjected to high deflections. Therefore, the 2D FE model was simulated in a *nonlinear static structural analysis* environment.

*Simulation outputs*: The behavior of the foot can be evaluated in two different modalities by plotting the *reaction force* at the fixed constraint against the platform *displacement* in the $y_{pf}$ direction (see Figure 4) or against the foot *rotation*. The *foot rotation* is calculated by using two virtual markers (**G** and **M** in Figure 8). The positions of the virtual markers were chosen considering the marker-set protocol [21,25]. Since the shank was fixed, the **J** marker was considered as (0,0).

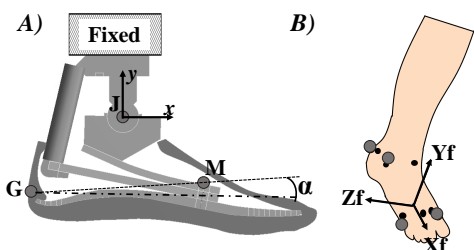

**Figure 8.** The foot rotation $\alpha$ is calculated as the variation of the angle between the shank axis and the **GM** line. The **G** and **M** markers taken into account in (**A**) are referred to the reflective markers (grey circles in (**B**)) considered by Leardini et al. [25].

The displacements of the virtual markers (given in the global $x$–$y$ reference) are given as direct output of the simulation and they can be considered as functions of the platform displacement $v_{pf}$ in the vertical direction $y_{pf}$ of the same platform (see Figure 4). Therefore, the displacements of the G marker in the global $x$- and $y$-directions are

$$u_G = u_G(y_{pf}) \tag{1}$$

$$v_G = v_G(y_{pf}) \tag{2}$$

The displacement of the M marker in the global $x$- and $y$-directions are

$$u_M = u_M(y_{pf}) \tag{3}$$

$$v_M = v_M(y_{pf}) \tag{4}$$

If $(x_{G0}; y_{G0})$ and $(x_{M0}; y_{M0})$ are the initial positions of G and M markers, referred to J $(0;0)$, the coordinates of G and M are:

$$x_G(y_{pf}) = x_{G0} + u_G(y_{pf}) \tag{5}$$

$$y_G(y_{pf}) = y_{G0} + v_G(y_{pf}) \tag{6}$$

$$x_M(y_{pf}) = x_{M0} + u_M(y_{pf}) \tag{7}$$

$$y_M(y_{pf}) = y_{M0} + v_M(y_{pf}) \tag{8}$$

The foot rotation $\Delta\alpha$ is calculated as the variation of the angle between the GM line and the horizontal direction (Figure 9). The initial angle $\alpha_0$ between the GM line and the horizontal direction and the rotation $\Delta\alpha$ of the foot are calculated as:

$$\alpha_0 = \arctan\left(\frac{y_{0G} - y_{0M}}{x_{0G} - x_{0M}}\right) \tag{9}$$

$$\Delta\alpha = \alpha - \alpha_0 \tag{10}$$

In Equation (10), $\alpha$ is determined as

$$\alpha = \arctan\left(\frac{y_G - y_M}{x_G - x_M}\right) \tag{11}$$

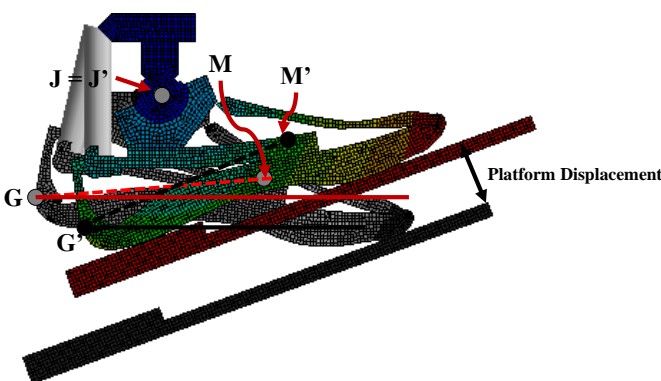

**Figure 9.** Virtual markers displacement during static dorsiflexion test. Platform displacement at 0 (black and white image) mm vs. platform displacement at 50 mm.

3.1.2. Material Properties Optimization: 3D FE Model

With the 2D FE model, the profile of the elastic elements and the general geometry of the foot prosthesis in the sagittal plane were defined. The 3D CAD model for the 3D FE model was built upon the geometry obtained in the previous step of the *design phase* methodology (Section 3.1.1). The elastic elements of ESR feet, in general, are laminate composite and they are built by stacking, in sequence, layers of CFRP (carbon fiber-reinforced plastic) or GFRP (glass fiber-reinforced plastic) prepregs, which are sheets of pre-impregnated fibers (generally pre-impregnated with resin) and they can be unidirectional or woven. The stiffness of each of the elastic elements of the ESR feet is defined by lamination sequence, type, number, orientation and order of each layer. For MyFlex-$\gamma$, unidirectional and woven CFRP prepregs were used. The lamination sequence of CFRP prepregs layers was optimized to reach the targeted stiffness, thus meeting the biomechanical requirements defined in Section 2.4. The elastic properties of the CFRP prepregs used are gathered in Table 5.

**Table 5.** Orthotropic elasticity of CFRP prepregs used to manufacture the upper blade, middle blade and lower blade.

| Type | Gramm. | Thick. | $E_1$ | $E_2$ | $E_3$ | $G_{12}$ | $G_{23}$ | $G_{13}$ | $\epsilon_{12}$ | $\epsilon_{23}$ | $\epsilon_{13}$ |
|------|--------|--------|-------|-------|-------|----------|----------|----------|------------------|------------------|------------------|
|      | g/m$^2$ | mm    | GPa   | GPa   | GPa   | GPa      | GPa      | GPa      | -                | -                | -                |
| UD   | 150    | 0.151  | 112.5 | 7.4   | 7.4   | 4.3      | 2.6      | 4.3      | 0.33             | 0.44             | 0.33             |
| UD   | 250    | 0.251  | 112.5 | 7.4   | 7.4   | 4.3      | 2.6      | 4.3      | 0.33             | 0.44             | 0.33             |
| W    | 200    | 0.234  | 61.3  | 61.3  | 6.9   | 3.3      | 3.3      | 2.7      | 0.04             | 0.30             | 0.30             |

*Loads and constraints*: The 3D FE model is intended to be validated with a mechanical test on a physical prototype of a foot prosthesis. In the ISO 10328 standard, the foot is loaded with an inclined platform, which means two inclined actuators are required, one for the heel and one for the toe, to compress the foot prosthesis. A dedicated test setup was designed and built (Figure 10) to avoid the necessity of two inclined actuators. The ISO 10328-equivalent test setup was characterized by a vertical piston that pushed a platform upward. The relative inclination between the foot and the platform was created by two different adapters that inclined the foot backwards by 15° and forwards by 20°. The platform, free to move along the longitudinal direction of the foot, compressed the foot prosthesis, fixed at the top. A vertical displacement was imposed to the platform to compress the foot at the heel and at the toe in two different tests to simulate the *ground*

*reaction forces* at both the early and the late stance, respectively. The total displacement was 10 mm and 50 mm at the heel and the toe load conditions, respectively, because of the different range of motion of the foot during plantarflexion and dorsiflexion. For the toe load, the platform was moved upward linearly at a rate from 3 mm/s to 4 mm/s. The platform moved linearly along the vertical direction for the heel load to compress the foot at a rate from 0.6 mm/s to 0.8 mm/s. These rate values, both for the plantarflexion and dorsiflexion tests, were chosen for a reason of convergence of the simulations. It was verified that the results did not change if the dorsiflexion simulation was performed at a rate of 3 mm/s or at a rate of 4 mm/s. The exact same applies to the plantarflexion simulation. This is justified by the fact that the simulation was still in rate values under static conditions.

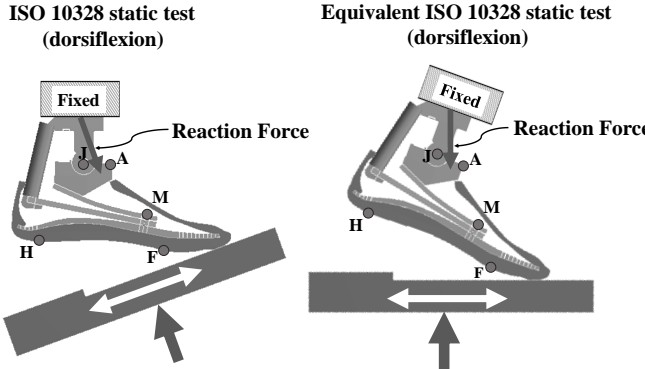

**Figure 10.** The ISO 10328 dorsiflexion static test compared to the equivalent test with vertical actuator.

The results from the *design phase* FEAs can be plotted as *reaction force* against the *rotation*, or *reaction force* against the platform *displacement*.

*Components modeling*: The 3D CAD Model was imported into ANSYS by setting *3D* as the analysis type. All isotropic parts were provided as solid to ANSYS, while composite parts were surfaces—Figure 11. All parts involved in contacts were modeled as flexible elements (platform, foot shell, lower blade, middle blade, upper blade, ankle frame and spring holder), while the parts connected with joints were modeled as rigid bodies (link and tube connector)—see Figures 12 and 13.

Based on the results obtained from the 2D FE model in the previous step, the initial values of thicknesses and elastic properties of the composite parts were predefined and used as a reference to define the lamination sequences of layers of CFRP prepregs. Then, the types, the numbers, the orientations and the order of the layers of CFRP prepregs were changed until the targeted thickness and elastic properties (calculated with the *Classical Theory of Laminates*) were reached.

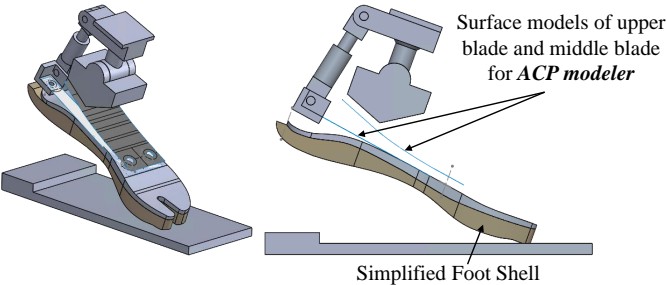

**Figure 11.** Isotropic parts were provided as solid, while composite components were surfaces. The Össur lower blade was assumed as isotropic with approximated properties. The foot shell was simplified considering the areas underneath the lower blade only.

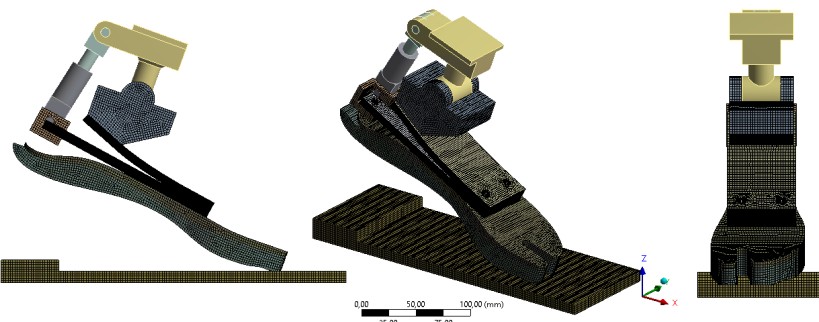

**Figure 12.** Mesh modeling of the components considered flexible in the analysis. The elastic elements, including the lower blade and the foot shell, were mainly modeled with SOLID186 elements (quadratic behavior). Ankle frame, spring holder and platform were modeled with SOLID185 elements (linear behavior).

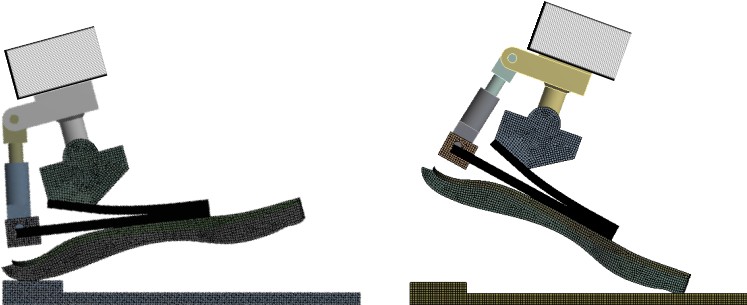

**Figure 13.** The 3D FE models for plantarflexion and dorsiflexion tests.

The 3D CAD model of the foot prosthesis was provided to ANSYS in *Geometry*. The lamination sequences of the composite parts were modeled inside *ANSYS Composite PrePost* (*ACP*). The main fiber direction for each of the composite parts is depicted in Figure 14 and it corresponds to the longitudinal direction of the foot prosthesis. Then, the composite parts were exported from *ACP* to *Static Structural modeler* as solid, with the final thicknesses.

*Mesh modeling*: The flexible components (platform, foot shell, lower blade, middle blade, upper blade, ankle blade and spring holder) were meshed. The elastic elements, including the lower blade and the foot shell, were mainly modeled with SOLID186 elements, which were 20-node elements with 3 degrees of freedom at each node and with quadratic behavior. SOLID187 elements (10-node elements with 3 degrees of freedom per node) were used for irregular areas. Ankle frame, spring holder and platform were modeled with SOLID185 elements, 8-node elements with 3 degrees of freedom at each node and linear behavior. The final 3D FE model of MyFlex-$\gamma$ consisted of around 1,800,000 degrees of freedom (600,000 nodes) (Figures 12 and 13).

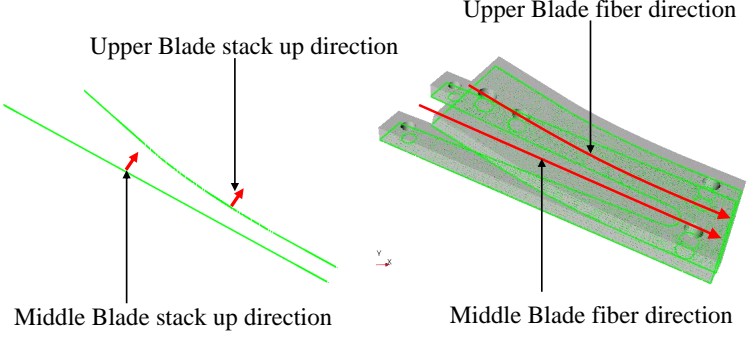

**Figure 14.** Modeling of the composite parts. The lower blade was considered to be isotropic with elastic properties close to Össur Pro-Flex Pivot's sole blade.

*Contact modeling*: As in the 2D FE model, for parts that bend to each other with slight sliding, the contacts were modeled as *frictional* using *pure penalty formulation* and 0.1 was set as the *normal stiffness factor* (the factor was set to 1 for normal dominated conditions). For contacts that could be modeled as glued/bonded, the *bonded* contact type was used with *augmented Lagrange formulation* and 1 was set as the *normal stiffness factor*. In MyFlex-γ, the upper blade, the middle blade and the lower blade bend to each other with slight sliding when the foot is loaded. The contacts between the upper blade and the middle blade and between the middle blade and the lower blade were modeled as *frictional contact* using *pure penalty formulation*. The contact between the ankle frame bottom surface and the upper blade top surface was modeled as *bonded* to simplify the simulation, using *augmented Lagrange formulation*. The contacts' properties used in the present 3D model are summarized in Table 6.

**Table 6.** Contacts' properties. See also Figure 15. AF = ankle frame; UB = upper blade; MB = middle blade; LB = lower blade; SH = spring holder; TC = tube connector.

| Surface 1 | Surface 2 | Type | Formulation | Frict. Coeff. | Norm. Stiff. Fact. |
|-----------|-----------|------|-------------|---------------|--------------------|
| AF top | UB bottom | bonded | augm.Lagrange | - | 1.00 |
| UB bottom | MB top | frictional | pure penalty | 0.20 | 0.01 |
| MB bottom | LB top | frictional | pure penalty | 0.20 | 0.01 |
| MB bottom | SH top | frictional | pure penalty | 0.20 | 0.01 |

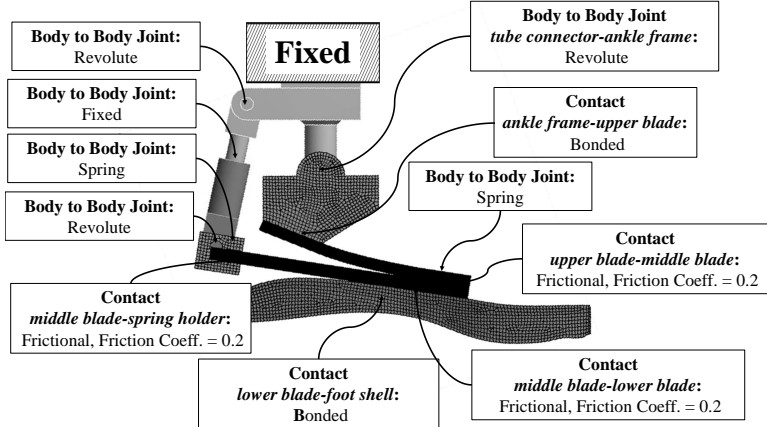

**Figure 15.** Joint and contact modeling in the 3D FE model. See also Table 6.

*Bolt modeling*: In MyFlex-γ, the elastic elements were joined together with two preloaded M8 bolts. In addition, the middle blade and the spring holder were joined with preloaded M6 screws. The screws were modeled as preloaded *spring joints* (ANSYS; COMBIN14) and with specific axial stiffness and pretensions that corresponded to M8 (front bolts) and M6 (spring holder bolts). The extremities of the springs were applied, as *remote attachments*, on the portions of the upper-blade top surface and on the bottom of the lower-blade surface to simulate the washers' section.

*Loads and constraints*: The FEA was carried out in a nonlinear static structural analysis by enabling the *large deflection* option. The *vertical displacement* was imposed on the platform, free to move along the x-direction and fixed along y (transverse)-direction (see Figures 10 and 13). The platform compressed the ESR foot and the reaction force was measured where the fixed support constraint was set.

## 3.2. Mechanical Test Phase

In the equivalent test setup, a hydraulic press machine (INSTRON 8033) was used as an actuator to compress the physical prototype of the foot prosthesis. The relative inclination between the foot and the platform was obtained by specifically designed adapters (Figure 16). As in the 3D FE model of the *design phase* (Section 3.1.2), the platform

was moved under displacement control; it moved upwards linearly with the same rate used in the *design phase*, i.e., 0.6–0.8 mm/s in the plantarflexion test and 3–4 mm/s in the dorsiflexion test, to replicate the same conditions. Thanks to a linear guide, the platform that pushed the foot prosthesis upward was free to move along the longitudinal direction of the same foot prosthesis. The platform was mounted on the linear guide, attached to a frame that could freely move when the hydraulic piston was actuated. The results of the mechanical test can be given as *displacement–reaction force*, where the displacement was the vertical motion of the platform and the reaction force was measured at the top of the inclination adapter with a load cell. By applying markers to the prosthesis of the foot, the rotations could also be determined; therefore, the stiffness curve of the prosthesis could be plotted as *rotation–reaction force*. Knowing, from the simulations, the ratio of the platform displacement and rotation of the foot, the results from the mechanical tests could be plotted as *rotation–reaction force* even without the use of markers.

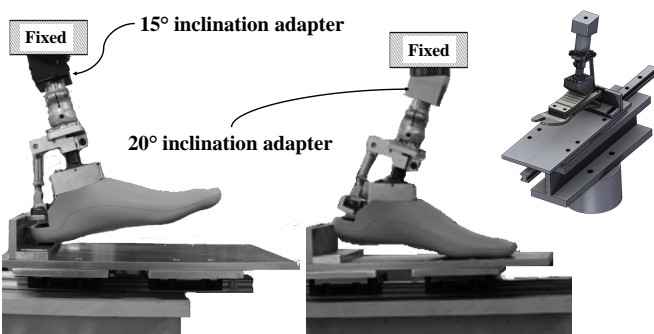

**Figure 16.** ISO 1328 equivalent static tests: actual plantarflexion and dorsiflexion tests and 3D Cad model of the test set up, in plantarflexion test configuration.

### 3.3. Functionality Verification Phase

The functionality verification of the foot prosthesis was conducted in two different modalities, both through transient structural FEAs. The first modality was a simplified dynamic test inspired by the cyclic tests from ISO 10328 (Section 3.3.2). The cyclic tests proposed in ISO 10328 relate to normal walking activities where loads occur regularly with each step. In the second modality, the roll-over task of the foot prosthesis was simulated and it is inspired by the dynamic tests from ISO 22675 and ISO/TS 16955 (Section 3.3.3).

#### 3.3.1. Two-Dimensional FE Model Adjustment

The 2D FE models used to simulate the two dynamic tests were built upon the 2D FE model developed in Section 3.1.1. The 2D FE model was characterized by speeding up the simulations in the case of geometry optimization, carried out by changing specific geometric parameters. Nevertheless, two-dimensional FE models are less precise than three-dimensional FE models. The first reason is, in the 2D FE model, the width of each part cannot be set as variable, as depicted in Figure 17. The second reason is geometric and concerns the holes, which are not considered in the 2D FE model, which is depicted again in Figure 17.

The third reason regards the material properties. The orientation of the fibers in composite materials is fundamental, in terms of both stiffness and strength. In the 3D FE model, the fibers were modeled following the curvatures of the components (middle blade and upper blade in MyFlex-$\gamma$). The same situation did not occur in the 2D FE model. Therefore, the 2D FE model built as described in Section 3.1.1 was simulated again. The widths of the elastic parts were slightly adjusted to obtain the same *displacement force* curves obtained in the static mechanical test (Section 3.2).

**2D FE Model**
*constant width for each part*
**3D FE Model**

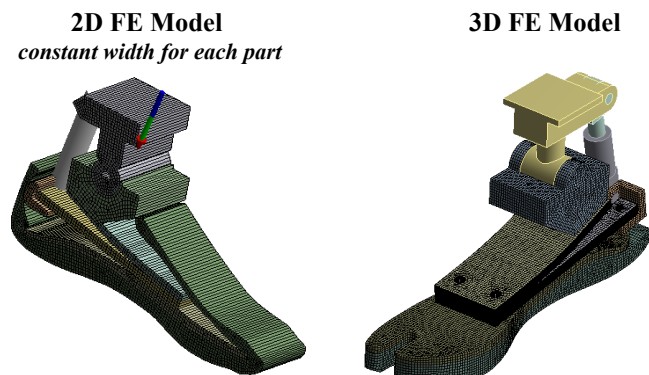

**Figure 17.** The width of each part in the 2D FE model was constant and the holes were not considered.

### 3.3.2. ISO 10328 Cyclic Tests

The re-adapted 2D FE model of the foot prosthesis was used to simulate the cyclic tests proposed in the ISO 10328 standard. Constraints, contacts, mesh modeling, joints, materials and width (dimension in the transverse direction) were the same for the foot prosthesis, while the load conditions were changed. The foot was loaded at the heel and forefoot with two platforms (Figure 18). The two platforms compressed the foot prosthesis with two forces (Figure 19), simulating the ground reaction forces during the gait cycle. The heel and the forefoot platforms were inclined by −15° and 20°, respectively. An initial space between the forefoot platform and the foot prosthesis was given (Figure 18). The forefoot touched the front platform when the forefoot force started to increase from 0 N (around 22% of the gait cycle, Figure 19). The first contact between the forefoot and forefoot platform simulated the toe strike. By considering the values of the *ground reaction forces* specified in Section 2 and the 60 kg weight category of users, the heel force peak was 764 N, while the forefoot force was 635 N. The same virtual markers used in Section 3.1.1 were used to calculate the rotation of the ankle–foot. For the dynamic analysis, an entire gait cycle of 1 s was considered.

**ISO 10328 dynamic test**

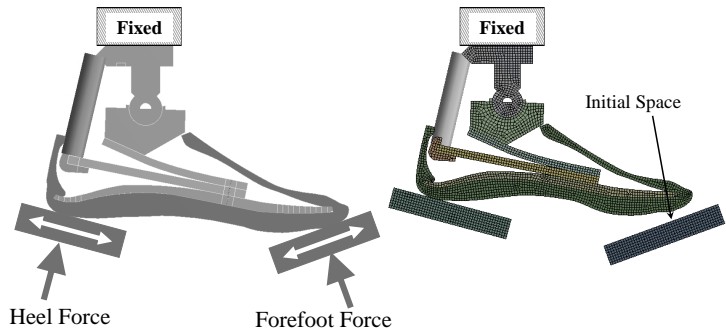

**Figure 18.** ISO 10328 cyclic test configuration: the heel platform was inclined at −15°, while the forefoot platform had an inclination of 20°. The heel and the forefoot forces followed the paths given in Figure 19.

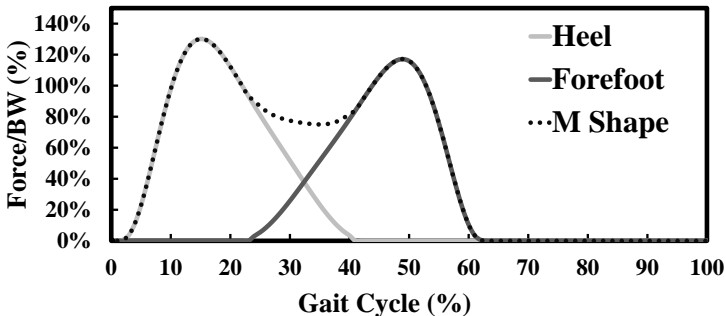

**Figure 19.** M-shaped force: the qualitative behavior of the heel and the forefoot forces are based on ISO 10328 dynamic test, while the maximum values (130% of the body weight for the heel force and 108% for the forefoot force) are from literature—see the *Biomechanical Requirements* in Section 2.4. In the present work, a 60 kg body weight was used; therefore the peaks were 764 N and 635 N. The forces were imposed as shown in Figure 18.

### 3.3.3. ISO 22675–ISO/TS 16955 Dynamic Test

The re-adapted 2D FE model of the foot prosthesis was used to build the present model to simulate the ISO 22675–ISO/TS 16955 dynamic/roll-over test. Other parts were added, such as the shank, the thigh and the tilting table (Figure 20). The contacts, mesh modeling, joints, materials and width (dimension in the transverse direction) were the same used in the first step of the *design phase*. The connection between the shank and the ankle–foot system was modeled as a bonded contact. The contact between the knee portion of the thigh and the knee portion of the shank was modeled in the FE model as a *no-separation* contact to create the knee joint. The thigh remained vertical and was free to move along the vertical direction only. The tilting table was rotated according to ISO 22675–ISO/TS 16955, while the *M-shaped force* was imposed from the top of the thigh (Figure 20). The angle between the foot and the tilting table ranged from −20° to 40° (Figure 20). The rotation of the foot during the roll-over test was calculated using the virtual markers G, M, J and K. First of all, the initial coordinates of each marker were determined from the CAD model as G ($x_{G0};y_{G0}$), M ($x_{M0};y_{M0}$), J ($x_{J0};y_{J0}$) and K ($x_{K0};y_{K0}$). The displacements of the virtual markers during the roll-over test were direct outputs of the simulation—$u_G$, $u_M$, $u_J$ and $u_K$, in the x-direction, and $v_G$, $v_M$, $v_J$ and $v_K$, in the y-direction. The coordinates of the markers during the roll-over test were: G($x_{G0} + u_G;y_{G0} + v_G$), M($x_{M0} + u_M;y_{M0} + v_M$), J($x_{J0} + u_J;y_{J0} + v_J$) and K($x_{K0} + u_K;y_{K0} + v_K$). The rotation $\Delta\alpha$ of the foot was given as the variation of the angle between the shank axis and the GM line. The angle $\alpha$ between the shank axis and the GM line was given as follows:

$$\alpha = \alpha_s - \alpha_{GM} \tag{12}$$

where $\alpha_s$ is the angle between the shank axis and the horizontal direction, calculated as

$$\alpha_s = \arctan\left(\frac{(y_{K0} + v_K) - (y_{J0} + v_J)}{(x_{K0} + u_K) - (x_{J0} + u_J)}\right) \tag{13}$$

whereas $\alpha_{GM}$ is the angle between the GM line and the horizontal direction, which is calculated as follows:

$$\alpha_{GM} = \arctan\left(\frac{(y_{G0} + v_G) - (y_{M0} + v_M)}{(x_{G0} + u_G) - (x_{M0} + u_M)}\right) \tag{14}$$

The rotation of the foot was then calculated in the following way:

$$\Delta\alpha = \alpha - \alpha_0 \tag{15}$$

with $\alpha_0$ as

$$\alpha_0 = \alpha_{s0} - \alpha_{GM0} \tag{16}$$

where $\alpha_{s0}$ and $\alpha_{GM0}$ are the initial angle between the shank axis and the horizontal direction and the initial angle between the GM line and the horizontal line, respectively. For the dynamic analysis, an entire gait cycle of 1 s was considered.

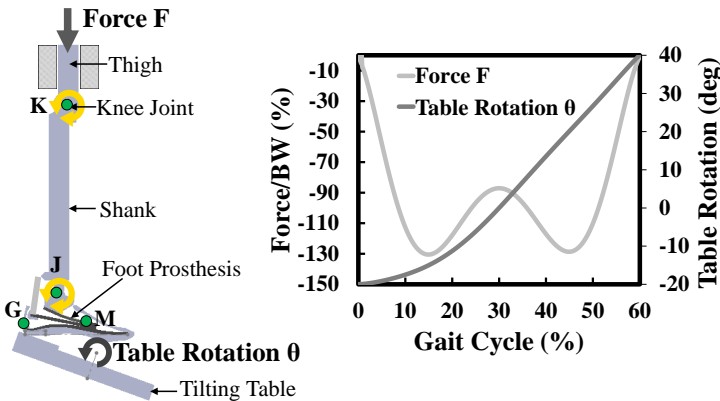

**Figure 20.** ISO–22675: the tilting table simulates the relative rotation between the ground and the thigh. The force F depends on the weight category and the maximum value was 130% of the body weight (BW). For the 60 kg weight category, the maximum value of the force F was 764 N. The tilting table ranged from $-20°$ to $40°$.

## 4. Results and Discussion

The results obtained from the *design*, the *mechanical test* and the *functional verification phases* are presented and discussed in the following sections. In addition, the comparison of the time elapsed to carry out the simulations is presented.

### 4.1. Design Phase Results

#### 4.1.1. Geometry Optimization: 2D FE Model Results

Several combinations of the parameters listed in Table 3 and depicted in Figure 5 were used to carry out more than 1000 simulations. From these 1000+ simulations, the final and pre-optimized values of the parameters were obtained and are listed in Table 7. The tabulated values are the final values from the 2D model for the weight category that was chosen to be developed (60 kg), while they are not the final values for the 3D model. In fact, of these values, the values of the thickness of the two blades in the 3D model change based on their stack-up sequence (which is defined in the 3D model, during the optimization of the properties of the materials).

**Table 7.** Final geometric parameters that define the profile shape of MyFlex-$\gamma$ elastic elements, for the 60 kg weight category. See Figure 5.

| Parameter | Final Value |
| --- | --- |
| upper blade thickness $UB_t$ (mm) | 6.80 |
| upper blade curvature 1 $UB\ c_1$ (deg) | 2.00 |
| upper blade curvature 2 $UB\ c_2$ (deg) | 2.00 |
| upper blade curvature 3 $UB\ c_3$ (deg) | 2.20 |
| upper blade curvature 4 $UB\ c_4$ (deg) | 2.50 |
| upper blade curvature 5 $UB\ c_5$ (deg) | 4.20 |
| middle blade length $MB_L$ (mm) | 163 |
| middle blade thickness $MB_t$ (mm) | 7.60 |

The choice of the final parameters was based on the *rotation–reaction force*. However, different configurations of the geometry could give similar results in terms of stiffness. Thus, stress and approximated final weight of the device were also used as evaluation parameters. However, among the 1000+ simulations, a configuration was considered optimal if the curve *foot rotation–reaction force* fell into the optimal area, highlighted in

the graphs in dark grey, as the intersection of the two lighter grey areas in Figure 21. These areas, both for the dorsiflexion and for the plantarflexion, were defined considering the biomechanical requirements previously reported in Section 2.4. Among the 1000+ combinations, the results from one of the non-optimal combination were taken into account to be compared with the results coming from the final combination of parameters that was considered pre-optimized. The dashed curve falls only partially into the optimal plantarflexion area (Figure 21), although the same combination of parameters already gives a curve that falls into the optimal dorsiflexion area. On the other hand, the black straight line is the curve that describes the stiffness of the combination of parameters considered pre-optimized. By varying the combinations of the parameters, similar stiffness curves can be obtained in dorsiflexion and different stiffness curves in plantarflexion or vice versa. This situation depends on the configuration of the elastic elements of each foot prosthesis to be optimized. In this application, one or more combinations of geometric parameters were considered to be those sought for biomechanical objectives if the stiffness curves fell within the ranges of rotations and loads defined in Section 2.4. However, for future works, exploiting the proposed methodology, optimization functions can be implemented in such a way that the choice of the final configuration is not based only on the evaluation of the stiffness curves.

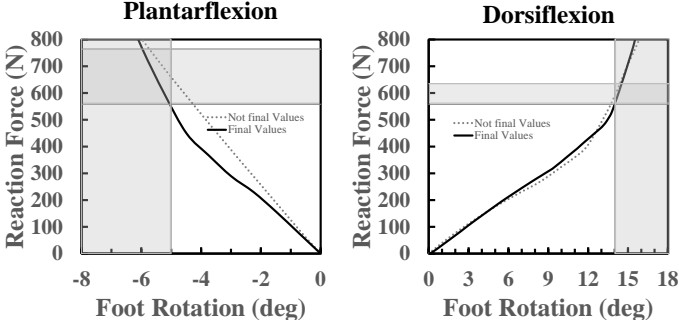

**Figure 21.** Results from the first stage of the *design phase*. The results from the pre-optimized configuration (Table 3) are compared to the results of a random configuration. The vertical shadow is the rotation range that the foot must have (values defined and specified in Section 2.4, "Biomechanical Requirements", i.e., between −5 degrees and −8 degrees in the plantarflexion and between 14 degrees and 18 degrees in the dorsiflexion). The horizontal shadow is the range of the ground reaction force (between 95% and 130% in plantarflexion and between 95% and 108% in dorsiflexion of the body weight of the user). The intersection area between the two shadows is the optimal area in which the curve of stiffness of the foot must fall, so that the foot rotates up to the degrees desired when subjected to ground reaction forces.

The stiffness curves, plotted as *foot rotation–reaction force* in Figure 21, are also provided as *platform displacement–reaction force*, as shown in Figure 22.

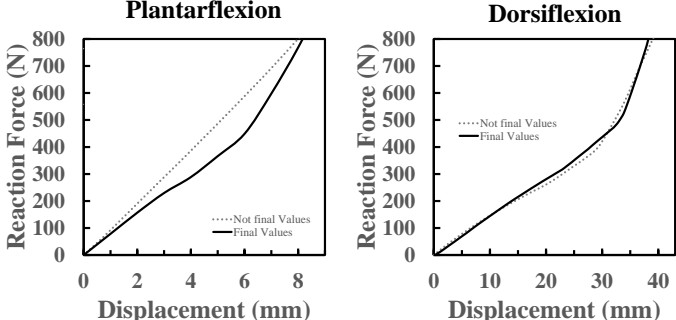

**Figure 22.** The stiffness curves presented as *foot rotation–reaction force* in Figure 21 can be also presented as *platform displacement–reaction force*.

### 4.1.2. Material Properties Optimization: 3D FE Model

The final lamination sequences of the upper blade and the middle blade were obtained after varying the types, the orientations and the numbers of layers of CFRP prepregs and are reported in Table 8.

**Table 8.** Final lamination sequences of the upper blade and the middle blade. The properties of the three types of CFRP prepregs are listed in Table 5. The directions of the fibers and the stack up for the upper blade and middle blade are depicted in Figure 14.

| Part | Type | Orientation (deg) | No. of Layers | Total Thickness (mm) |
|---|---|---|---|---|
| | Woven 200 g/m$^2$ | 0 | 3 | 0.702 |
| | Woven 200 g/m$^2$ | 45 | 2 | 0.468 |
| Upper blade | Unidir. 250 g/m$^2$ | 0 | 18 | 4.518 |
| | Woven 200 g/m$^2$ | 45 | 2 | 0.468 |
| | Woven 200 g/m$^2$ | 0 | 3 | 0.702 |
| | | | total = | 6.858 |
| | Woven 200 g/m$^2$ | 0 | 3 | 0.702 |
| | Unidir. 250 g/m$^2$ | 0 | 5 | 1.255 |
| | Unidir. 150 g/m$^2$ | 0 | 10 | 1.510 |
| Middle blade | Woven 200 g/m$^2$ | 0 | 3 | 0.702 |
| | Unidir. 150 g/m$^2$ | 0 | 10 | 1.510 |
| | Unidir. 250 g/m$^2$ | 0 | 5 | 1.255 |
| | Woven 200 g/m$^2$ | 0 | 3 | 0.702 |
| | | | total = | 7.636 |

This prototype was made with the aim of validating the two 2D and 3D models of the *design phase*. For the present application, no fiber orientation other than 0° and 45° was used, to simplify the manufacturing of the prototype. Other orientations, such as 30 degrees, may also be considered for further optimization. As can be seen in Table 8, 45° oriented fibers were only found in the upper blade, while, in the middle blade, they were absent. Based on the result from the point of view of stress/strength, fibers woven oriented at 45° led to too-high stress values when the prosthesis was subjected to operating loads and this can be justified by the shape of the middle blade, shown in Figure 23. The two parts into which the middle blade was divided into its fork shape were dimensionally too narrow; further, 45° oriented fibers reduced the strength of the entire laminate.

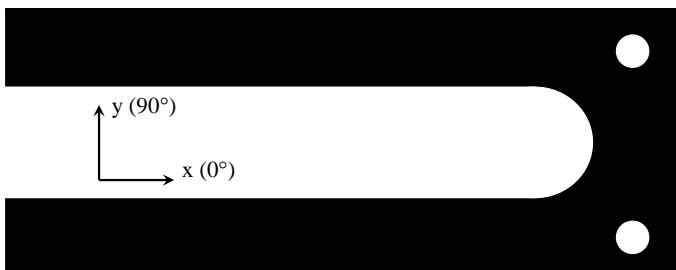

**Figure 23.** Shape of the middle blade of MyFlex-γ.

The stiffness in dorsiflexion and plantarflexion of the foot prosthesis (modeled with the final lamination sequences) are presented as *foot rotation–reaction force* in Figure 24 and as *platform displacement–reaction force* in Figure 25.

Considering the strength properties of the materials used to realize the elastic components (Table 9), the stress–strength ratio analysis for these components was carried out. The Tsai–Wu criterion was used as the assessment criterion of the composite parts (the upper blade and the middle blade). When the foot was loaded with the 220% of the weight

category, the *inverse reserve factor* was under 0.5 (Figure 26) in the most critical area. An *inverse reserve factor* of 0.5 corresponded to a *safety factor* of 2.

**Table 9.** Orthotropic strength of CFRP prepregs used to manufacture the upper blade, middle blade and lower blade. $T_1$, $T_2$ and $T_3$ are the tensile strength; $C_1$, $C_2$ and $C_3$ are the compressive strength; $S_{12}$, $S_{23}$ and $S_{13}$ are the shear strength.

| Type | Gramm. | Thick. | $T_1$ | $T_2$ | $T_3$ | $C_1$ | $C_2$ | $C_3$ | $S_{12}$ | $S_{23}$ | $S_{13}$ |
|------|--------|--------|-------|-------|-------|-------|-------|-------|----------|----------|----------|
|      | g/m$^2$ | mm | MPa | MPa | MPa | MPa | MPa | MPa | MPa | MPa | MPa |
| UD | 150 | 0.151 | 2200 | 29 | 29 | −1082 | −100 | −100 | 60 | 30 | 60 |
| UD | 250 | 0.251 | 2200 | 29 | 29 | −1082 | −100 | −100 | 60 | 30 | 60 |
| W | 200 | 0.234 | 805 | 805 | 50 | −509 | −509 | −170 | 125 | 65 | 65 |

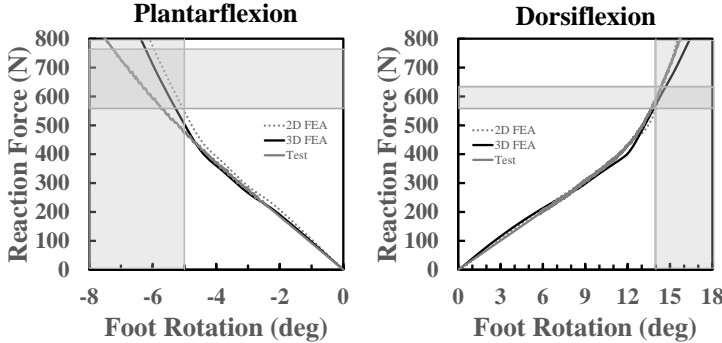

**Figure 24.** Comparisons among the static plantarflexion and dorsiflexion 2D FEAs and 3D FEAs and the mechanical tests on the physical prototype with stiffness curves givens as *reaction force–rotation*. Again, the vertical shadow is the rotation range that the foot must have (values defined and specified in Section 2.4, "Biomechanical Requirements", i.e., between −5 degrees and −8 degrees in the plantarflexion and between 14 degrees and 18 degrees in the dorsiflexion). The horizontal shadow is the range of the ground reaction force (between 95% and 130% in plantarflexion and between 95% and 108% in dorsiflexion of the body weight of the user). The intersection area between the two shadows is the optimal area in which the curve of stiffness of the foot must fall, so that the foot rotates up to the degrees desired when subjected to ground reaction forces.

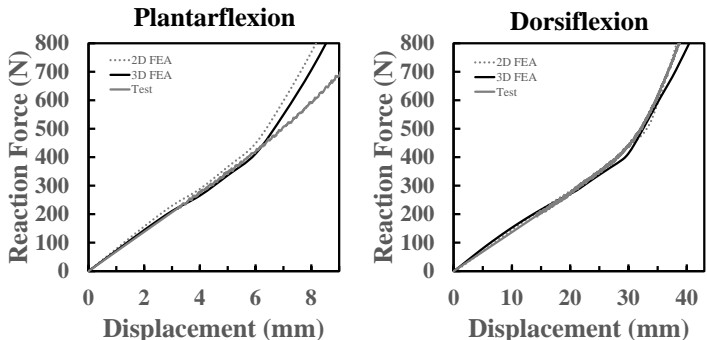

**Figure 25.** Comparisons among the static plantarflexion and dorsiflexion 2D FEAs and 3D FEAs and the mechanical tests on the physical prototype with stiffness curves given as *reaction force–displacement*.

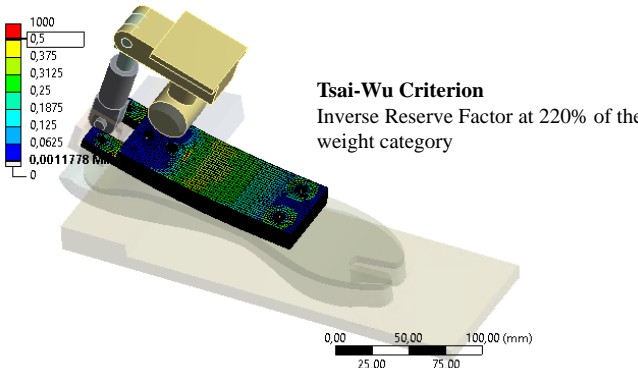

**Figure 26.** Upper blade and middle blade evaluated with the Tsai–Wu criterion; the most critical areas, when the foot was loaded with the 220% of body weight of intended users, presented an *inverse reserve factor* of 0.5, which means a *safety factor* of 2.

### 4.2. ISO 10328 Equivalent Static Test Results

Once the stiffness optimization of its elastic components was performed, a prototype of MyFlex-γ was built. The composite elements were manufactured using the lamination sequences for the upper blade and the middle blade listed in Table 8. In addition, the other components were designed with the same safety factor. The assembled physical prototype was tested with the ISO-10328 equivalent static test setup, as presented in Section 2.3. The results are plotted as *rotation–reaction force* in Figure 24 and as *displacement–reaction force* in Figure 25.

The dorsiflexion curves, which are reported as *rotation–reaction force* (Figure 24), or as *displacement–reaction force* (Figure 25), are quite similar. Therefore, it can be said that both the 2D FE and 3D FE models were validated with the mechanical tests carried out on the physical prototype of MyFlex-γ.

If the curves of dorsiflexion are very similar to each other, the same cannot be said for those of plantarflexion. In fact, 2D FEA and 3D FEA curves diverge from the test curve obtained for the physical prototype. The reasons for the divergence could be the following: (i) the not-perfect mounting of the foot prosthesis on the test set-up; (ii) the not perfect CAD and FE modeling of the foot shell; (iii) manufacturing defects that can cause slight variations in the geometry of the physical prototype; (iv) the joints, such as the ankle joint, hinge joints between the link and tube connector, and between the link and spring holder, modeled without friction—thus, approximated; (v) the heel of the foot shell not fully covered by the platform, as depicted in Figure 27.

Interference between *spring holder* and the *foot shell*

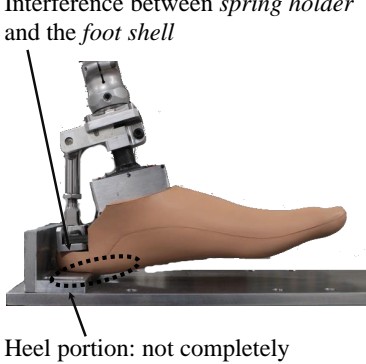

Heel portion: not completely covered by the heel platform

**Figure 27.** Plantarflexion test: the heel of the foot shell was not completely covered by the platform.

The plantarflexion *displacement–reaction force* curve obtained from the 3D FEA resulted stiffer than the curve obtained from the mechanical test after around 6–7 mm of the vertical displacement of the platform. After 6–7 mm of vertical displacement, the heel was not

fully supported by the heel platform. The vertical support caused this situation (Figure 27), which was not considered in the 3D FEA. During the heel test, the contact area between the heel and the platform shifted forward. The distance between the point of application of the force and the center of the ankle joint was reduced. This condition did not occur in the mechanical test after 6–7 mm; the contact between the foot and the heel platform did not evolve as it should have. With the same torsional stiffness of the foot prosthesis in the sagittal plane, the force necessary to generate a plantarflexion was higher in the 3D FEA due to the shorter lever arm. It has to be specified that the platform was built according to standards. However, the space for the heel portion of the foot was not enough. For a more appropriate assessment of the plantarflexion stiffness, the heel platform (see Figure 27) should provide more space, or the vertical face should be removed. In the case of the dorsiflexion test, the test setup did not present a space issue. It is interesting to note that the two dorsiflexion stiffness curves are in good agreement. This condition should provide confidence in the 3D FE model presented in the *design phase* (Section 3.1.2).

### 4.3. Functionality Verification Results

#### 4.3.1. ISO 10328 Cyclic Tests

Figure 28 shows how the elastic elements were deflected when the foot was subjected to the ISO 10328 cyclic test.

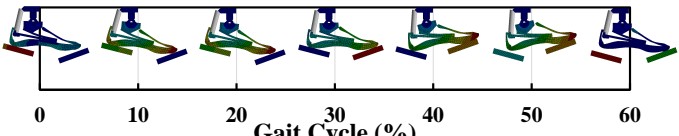

**Figure 28.** The deflection sequence of the 2D FE model of the prosthesis during ISO 10328 cyclic test.

The foot prosthesis had its maximum plantarflexion around −6.4° and maximum dorsiflexion around 15°. Both the values are comprised in the ranges specified in the *Requirements* section (Section 2.4), as also highlighted in Figure 29.

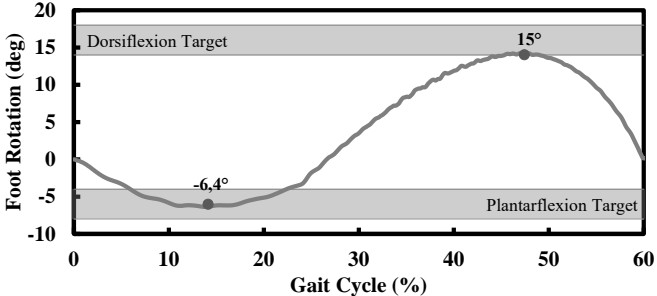

**Figure 29.** The rotation of the foot prosthesis when subjected to the ISO 10328 cyclic test.

#### 4.3.2. ISO 22675–ISO/TS 16955 Roll-over Test

The behavior of the of the leg prosthesis during ISO 22675 is shown in Figure 30. The maximum plantarflexion of MyFlex-γ was −5°, while the maximum dorsiflexion was 15°. The maximum plantarflexion obtained during *early stance* is comprised in the range of the targeted angles during normal walking. In addition, the maximum dorsiflexion during *mid-stance* is inside the aimed range (Figure 31).

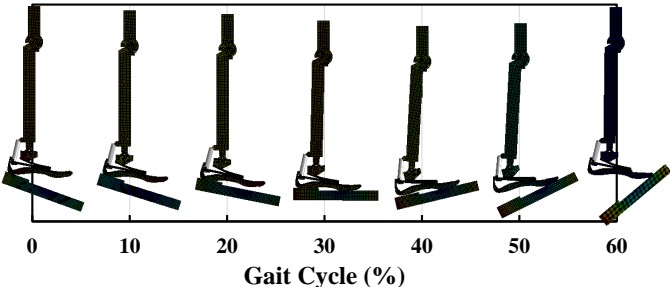

**Figure 30.** The lower leg prosthesis behavior when subjected to the ISO 22675 roll-over test.

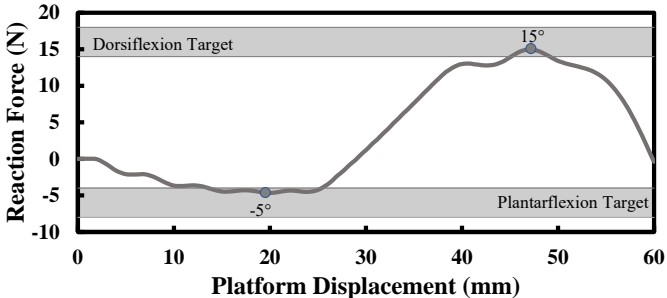

**Figure 31.** The rotation of the foot prosthesis when subjected to the ISO 22675 roll-over test.

### 4.3.3. Elapsed Time of Calculation Comparison: 2D FE Model vs. 3D FE Model

One of the advantages of the present methodology is to reduce the time of calculation during geometry optimization, where the effects of the variation in several geometric parameters are investigated. In previously published works on foot prostheses, geometry optimization has been carried out through 3D FE models [12,13]. The 2D FE model aims to provide a design tool to study the influence of the geometry on the foot prosthesis behavior, giving the result in a shorter time than performing 3D FEAs. The time elapsed to carry out a structural FEA depends on the computing power of the calculator used. However, in similar conditions (same computing power), the time necessary to perform the FEAs depends on several factors, such as the number of degrees of freedom of the FE models. In Table 10, the number of nodes and number of degrees of freedom from the 2D FE model (Section 3.1.1) and the 3D FE model (Section 3.1.2) are listed. In the present specific application, the 2D static FEA that simulated the dorsiflexion test required a time of calculation of 45 s, while the equivalent 3D FEA required a time of calculation of 4 h and 19 min.

**Table 10.** Number of nodes and number of degrees of freedom from the 2D FE model (Section 3.1.1) and the 3D FE model (Section 3.1.2).

| Step | Type of Simulation | Number of Nodes | Number of Deg. of Freedom |
|------|--------------------|-----------------|---------------------------|
| 1 | 2D Static | 10,000 | 20,000 |
| 2 | 3D Static | 600,000 | 1,800,000 |

### 5. Conclusions

A methodology based on finite element (FE) analysis (FEA) and experimental techniques for designing the elastic elements of prosthetic feet and their functionality verification is presented. The present methodology can be exploited for different purposes. It can be used to design new configurations of energy-storing and -releasing (ESR) feet for different weight categories. It can be applied as a tool to study new systems that can be added to an existing ESR foot to change its stiffness/damping properties. Furthermore, it can be used to study the behavior of active prosthetic feet where their working principle relies on actuators combined with the elastic foot. The methodology is subdivided into

three main phases: the *design phase*, the *validation phase* and the *functionality verification phase*. The *design phase* consists of two steps based on FEA. In the *validation phase*, the FE models are validated through a static test on physical prototypes. In the *functionality verification phase*, two standard dynamic tests from ISO 10328 and ISO 22675 are simulated. In the case study of the ESR foot MyFlex-$\gamma$, the static 2D FE model from the first step of the *design phase* resulted to be a reliable and fast-response tool to predict the influence of geometric parameters on the behavior of the ESR foot. In addition, the 2D FE model resulted to be a reliable instrument to find the optimal values of these geometric parameters. With the static 3D FE model (the second step of the *design phase*), it was possible to optimize the material properties of the composite elastic elements. By optimizing the geometry and the material properties, it was possible to obtain the correct stiffness of the foot prosthesis for a specific weight category of users, providing a device with a safety factor of 4. The stiffness curves obtained from the 3D model resulted comparable to the results from the ISO 10328 equivalent mechanical test (*validation phase*). Therefore, the FE models can be considered validated. The rotation angles of the foot obtained in the model based on the dynamic test ISO 10328 and the model based on the dynamic test ISO 22675 are within the target range of motion. The two dynamic tests can be considered alternatives to each other. They can be both considered helpful for a preliminary assessment of the prosthesis dynamic behavior before testing the prototype with amputee users.

**Author Contributions:** Conceptualization, J.T., T.M.B. and M.O.; methodology, J.T., T.M.B, M.P., M.L. and M.O.; validation, J.T., T.M.B. and M.L.; formal analysis, J.T.; investigation, J.T., T.M.B. and M.L.; data curation, J.T.; writing—original draft preparation, J.T.; writing—review and editing, J.T., T.M.B, M.P., M.L., M.O., R.C. and A.Z.; visualization, J.T. and A.Z.; supervision, M.O., R.C. and A.Z.; resources: M.O., A.Z. and M.P.; project administration, M.O., R.C. and A.Z.; funding acquisition, R.C. and A.Z. All authors have read and agreed to the published version of the manuscript.

**Funding:** This work was funded by the European Commission's Horizon 2020 Programme as part of the project MyLeg under grant no. 780871.

**Acknowledgments:** The authors want to thank Eleonora Sotgiu and Pietro Benincasa (MSc students at University of Bologna) for their fundamental contributions to the design and manufacturing of MyFlex-$\gamma$. The authors want also to thank Stefano Monti (DIN Technician at University of Bologna), Mauro and Lorenzo Sassatelli (Metal TIG S.r.l.—www.metaltig.it, Italy, access date: 1 January 2021) for their contributions to the realization of the prototypes.

**Conflicts of Interest:** The authors declare no conflict of interest. The funders had no role in the design of the study; in the collection, analyses, or interpretation of data; in the writing of the manuscript, or in the decision to publish the results.

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
