# Peer review of "Structural FEA-Based Design and Functionality Verification Methodology of Energy-Storing-and-Releasing Prosthetic Feet"

_applsci, doi:10.3390/app12010097_

Round 1
Reviewer 1 Report
The present work proposes a methodology that describes the process of designing prosthetic legs. The methodology consists of three main phases- the design phase, mechanical prototype testing phase, and functional verification phase.
A deep literature study has been made and the experience of other authors in the field of prosthesis design and in particular the use of FEA in the design process is indicated.
The detailed description when applying the finite element method conveys a very good impression. Thus, the article can be useful to a wide range of readers.
Creating and testing a physical prototype contributes to increasing the scientific value of the publication.
I recommend publishing the paper.
Author Response
Dear Sir/Ma’am,
Thank you for kindly reviewing the manuscript.
Reviewer 2 Report
This manuscript deals with the design, analysis, and experimental test of prosthetic feet. The Finite Element Analysis was used to design and predict the effective stiffness of prosthetic feet elastic elements. The results (geometry and layup of composite laminate) obtained from FEA have been used to manufacture elastic elements and experimentally validate the FE model. Overall, the manuscript includes many works, but it misses some essential details about the state of the optimization problem and sensitivity analysis of selected parameters to the obtained results.
- In the paper, the keyword „optimization“is mentioned many times, while it is not clear what the authors tried to minimize or maximize? What was the optimization function? The optimization problem is not formulated properly.
- After a comprehensive description of the methodology, the obtained results are presented superficially. The authors mentioned that it was performed more than thousands of iterations to get some results, but sensitivity analysis of selected parameters was not presented.
- In fig. 24 and 25 presented stiffness curves with specific but not final values of parameters. But what are those specific values and how do the curves look with final values? How did the authors decide which are the optimal parameters, if with other specific parameters (as shown in Fig. 24) the stiffness curve goes through the optimal region also? Looks that similar stiffness could be achieved and having a smaller thickness than final values (presented in Table 7) because in the final layup there are quite a lot of woven layers. Again, the criteria of the Optimization problem are not clear.
- As the results of strength analysis for composite laminates are presented, the strength properties of materials should be given. Also, Poisson ratios are missing.
- Why in the Upper blade the 45 deg orientation was needed and for the Middle blade not? There were some in-plane shear forces? What was the step for orientation degrees? It was only two options: 0 and 45 degrees, or it was analyzed any possible degree of orientation?
- The results of 2d and 3D analysis given in Fig. 24-28 should be combined for easier comparison. Also into the combined figures could be integrated results from tests (Fig. 30-31).
- The name of Table 8: Fig. ?? should be corrected.
Author Response
Dear Sir/Ma’am,
Thank you for kindly reviewing the manuscript.
Below you can find the response to the comments.
Point 1: In the paper, the keyword “optimization” is mentioned many times, while it is not clear what the authors tried to minimize or maximize? What was the optimization function? The optimization problem is not formulated properly.
Response 1: Stiffness curves, given as platform displacement - reaction force or foot rotation - reaction force, must be included in the optimal ranges chosen by the authors based also on human foot rotations for individuals with ambulation level K3 and K4. Each combination of geometric parameters provides a stiffness curve in dorsiflexion and one in plantarflexion. Although the effects of each of the geometric parameters on the overall stiffness of the foot prosthesis for other purposes have been studied, the detailed results of this study have not been presented, as the aim of this article is to present the design methodology so that it can be used by others in the future. Future users, using the procedure presented in the methodology, will be able to formulate their optimization function according to their objective. What has been done in general for this work is to consider the combination of geometric parameters that give two optimal stiffness curves (dorsiflexion and plantarflexion) according to the objectives chosen by the authors. For future work, the same methodology can for example be used for foot implants for users with less mobility than individuals with outpatient level K3 and K4, clearly choosing biomechanical constraint other than those presented in this work, which are specific to Myflex – γ. Or, as was done for the later version of Myflex – γ, the methodology presented was used to find a geometric parameter that can be varied in use to obtain a foot prosthesis with variable stiffness (Myflex – δ, a semi-active foot prosthesis with variable stiffness and developed upon Myflex - Myflex – γ, will be described in a future work citing this article).
Point 2: After a comprehensive description of the methodology, the obtained results are presented superficially. The authors mentioned that it was performed more than thousands of iterations to get some results, but sensitivity analysis of selected parameters was not presented.
Response 2: Several analyses and simulations have been made, but since the paper’s focus is on the design methodology and not on numerical validation, it has not been inserted to avoid weighing down the paper. If required, the author is available to integrate, even if a future and dedicated article is aimed to be written (citing the present paper). As also specified in the previous response, with the 2D model was also made a specific study in which it was studied the effect of each of the parameters on the overall stiffness of the foot quantifying this effect. However, such a study is not fundamental to a prototype built from scratch. It can be useful once the first prototype is made and it is clinically tested by the future user. Following the user feedback after the functional tests in the clinic, the study of the effects of each of the parameters can be exploited to make changes on the geometry in such a way that, not only does the foot rotate by exactly n degrees when subjected to specific GRF values, but also how the foot rotates from 0 to the target degrees, both in dorsiflexion and plantarflexion.
As mentioned also in the previous response, taking advantage of the study of the effects of each of the parameters, Myflex - δ a semi-active prosthesis with variable stiffness was made starting from Myflex – γ, and using the present methodology in the design phase, already knowing the objectives thanks to the results of the study of the effects of the parameters on the general stiffness of the foot.
Point 3: In fig. 24 and 25 presented stiffness curves with specific but not final values of parameters. But what are those specific values and how do the curves look with final values? How did the authors decide which are the optimal parameters, if with other specific parameters (as shown in Fig. 24) the stiffness curve goes through the optimal region also? Looks that similar stiffness could be achieved and having a smaller thickness than final values (presented in Table 7) because in the final layup there are quite a lot of woven layers. Again, the criteria of the Optimization problem are not clear.
Response 3: In Fig. 24 and 25 the authors have wanted to present an example of curves of stiffness not optimized, that is curves of stiffness that are not inside the objectives of Range of Motion of the ankle. In the revised version of the Manuscript, optimal 2D curves are added. There was an error in using the term "specific". The curves in Fig. 24 and Fig. 25 of the original manuscript were drawn by randomly choosing the results of an optimized parameter configuration. It is true that similar stiffness can be obtained with smaller thicknesses. However, the results of the FEM 3D analysis from the point of view of stress-strength led us to have the final layups presented, considering the loads that the foot prosthesis must undergo during its use. It is true that the stiffness of the foot is optimized on the sagittal plane, but it is also true that, since the prosthesis is not subject only to loads in the sagittal plane even when only loads are imposed in that plane due to the fact that the prosthesis is not symmetrical. For example, in the dorsiflexion test, the prosthesis is touched by the platform initially only by what the big toe should be (see Fig. 3); this situation generates in the foot a dorsiflexion but also rotations in the frontal plane and in the transverse plane. Using only unidirectional layers or reducing the number of woven, the strength of the foot would not be sufficient for the loads to which it would be subjected during use.
Point 4: As the results of strength analysis for composite laminates are presented, the strength properties of materials should be given. Also, Poisson ratios are missing.
Response 4: In the revised manuscript, strength properties and Poisson ratios have been added.
Point 5: Why in the Upper blade the 45 deg orientation was needed and for the Middle blade not? There were some in-plane shear forces? What was the step for orientation degrees? It was only two options: 0 and 45 degrees, or it was analyzed any possible degree of orientation?
Response 5: Other fibre orientations were also analysed. However, for simplicity of prototype production (authors cut all layers by hand), the choice was made to find the layups that give the desired stiffness by placing as a constraint in the design phase that of using only 0 and 45 degree orientations. However, those who want to use this proposed method can use other orientations. Concerning the Middle Blade, the choice not to put layers oriented at 45 degrees was dictated by the shape and size of the "fork" zone of the same component. Layers oriented to 45 degrees would have reduced its strength.
Point 6: The results of 2d and 3D analysis given in Fig. 24-28 should be combined for easier comparison. Also into the combined figures could be integrated results from tests (Fig. 30-31).
Response 6: In the revised manuscript, the results from 2D, 3D and test have been integrated.
Point 7: The name of Table 8: Fig. ?? should be corrected.
Response 7: In the revised manuscript, the problem has been fixed.
Reviewer 3 Report
In this manuscript, a methodology based on finite element analysis and experimental techniques for designing the elastic elements of prosthetic feet was presented. The author subdivided the whole design into three main phases, and explained each steps of each phase in detail. The ESR foot MyFlex-γ examples was used to illustrate and display the design steps. I think the work is meaningful.
There are some suggestions about the manuscript:
- Page 12 Line 317 and Page 15 Line 379: The author mentioned “the platform…moves…with a rate of 0.8 mm/s for the plantarflexion test and 3 mm/s for the dorsiflexion test.”. Please explain the basis for setting these two speed value.
- Page 16 Line 467: How to define “the optimal conditions”? It is suggested a more detailed explanation, so as to the Figure 24. In Figure 24, please explain the meaning of the vertical and horizontal shadows, and the meaning of overlapping areas of the shadows. How are the positions of these shadow areas obtained.
- There’s a little mistake in Page 20, the title of Table 8. “The directions of the fibers and the stack up for the upper blade and middle blade are depicted in Fig. ??.” Which figure does the author wanted to mention?
Author Response
Dear Sir/Ma’am,
Thank you for kindly reviewing the manuscript.
Below you can find the response to the comments.
Point 1: Page 12 Line 317 and Page 15 Line 379: The author mentioned “the platform…moves…with a rate of 0.8 mm/s for the plantarflexion test and 3 mm/s for the dorsiflexion test.”. Please explain the basis for setting these two speed value
Response 1: In the various simulations made during the design phase, the two tests were simulated by setting as rates 0.6-0.8 mm/s for plantarflexion and 3-4 mm/s for dorsiflexion. In the revised manuscript it has been modified. It has been verified that the results do not change if the dorsiflexion simulation is performed at a rate of 3 mm/s or at a rate of 4 mm/s. The exact same applies to the plantarflexion simulation. This is justified by the fact that the simulation is still in a rate values under static conditions. These values were chosen for a reason of convergence of the FEM simulations.
Point 2: Page 16 Line 467: How to define “the optimal conditions”? It is suggested a more detailed explanation, so as to the Figure 24. In Figure 24, please explain the meaning of the vertical and horizontal shadows, and the meaning of overlapping areas of the shadows. How are the positions of these shadow areas obtained.
Response 2: The vertical shadow is the rotation range that the foot must have (values defined and specified in Section 2.4, "Biomechanical Requirements": between -5 degrees and -8 degrees in the plantarflexion, between 14 degrees and 18 degrees in the dorsiflexion). Horizontal shadow is the range of the ground reaction force (between 95% and 130% in plantarflesion, between 95% and 108% in dorsiflexion of the body weight of the user). The intersection area between the two shadows is the optimal area in which the curve of stiffness of the foot must fall, to make that the foot rotates up to the degrees desired when subjected to ground reaction forces. In the revised manuscript these specifications were added to make the explanation clearer.
Point 3: There’s a little mistake in Page 20, the title of Table 8. “The directions of the fibers and the stack up for the upper blade and middle blade are depicted in Fig. ??.” Which figure does the author wanted to mention?
Response 3: The problem was fixed in the revised version of the manuscript.
Round 2
Reviewer 2 Report
The reviewer is fully satisfied with the response of the authors and improvements in the manuscript ("Structural FEA-based Design and Functionality Verification Methodology of Energy-Storing-and-Releasing Prosthetic Feet "). I confirm that the manuscript has been sufficiently improved to warrant publication in Applied Sciences.